# Phase-Inverted Copolymer Membrane for the Enhancement of Textile Supercapacitors

**DOI:** 10.3390/polym14163399

**Published:** 2022-08-19

**Authors:** Sheng Yong, Nicholas Hillier, Stephen Paul Beeby

**Affiliations:** 1Smart Electronic Materials & System Research Group, School of Electronics and Computer Science, University of Southampton, Southampton SO17 1BJ, UK; 2Energy Storage and Its Applications Centre of Doctoral Training, University of Southampton, Southampton SO17 1BJ, UK

**Keywords:** supercapacitor, e-textile, phase inversion membrane, energy storage device, polymer separator, carbon electrode

## Abstract

This paper presents a universal fabrication process for single-layer textile supercapacitors, independent of textile properties such as weave pattern, thickness and material. To achieve this, an engineered copolymer membrane was fabricated within these textiles with an automated screen printing, phase inversion and vacuum curing process. This membrane, together with the textile yarns, acts as a porous, flexible and mechanically durable separator. This process was applied to four textiles, including polyester, two polyester-cottons and silk. Carbon-based electrodes were subsequently deposited onto both sides of the textile to form the textile supercapacitors. These supercapacitors achieved a range of areal capacitances between 3.12 and 38.2 mF·cm^−2^, with energy densities between 0.279 and 0.681 mWh·cm^−3^ with average power densities of between 0.334 and 0.32 W·cm^−3^. This novel membrane facilitates the use of thinner textiles for single-layered textile supercapacitors without significantly sacrificing electrochemical performance and will enable future high energy density textile energy storage, from supercapacitors to batteries.

## 1. Introduction

The integration of electronics within textiles (e-textiles) has seen a plethora of applications unlocked for medical [1], defence [2] and consumer [3] wearable technologies. Increasingly though, the power management of these technologies has become a bottleneck for further designs. This requirement for power bridging (analogous to increasing the penetration of renewable energy onto electrical networks) or power averaging has seen a stark increase in research interest in textile supercapacitors (TSCs) and batteries. This is highlighted by an increase in publications of two orders of magnitude over the previous decade [4]. TSCs have been far more widely researched than other textile energy storage systems due to their increased energy and power density when compared to traditional capacitors [5] and their longer cycle lifetime and reliability when compared to batteries [6]. Similar to a traditional capacitor, a supercapacitor comprises two electrodes, an electrolyte and an electrically insulating but ionically conductive separator. To date, electrode materials for TSCs have seen extensive research with advances in conductivity [7], porosity [8], mechanical and chemical stability [9] and even functionalisation [10]. The electrolyte likewise has seen significant research with a lot of the focus on aqueous gel-based formulations (H_2_SO_4_ [11], H_3_PO_4_ [12], KOH [13]), but also some exciting explorations into the use of nanoparticle fillers [14]. Despite this, the separator layer has seen little advancement, with it often being realised simply as an extra layer of separator paper sandwiched between textile electrodes [15] or as an additional film layer [16] in a sandwich-like configuration. These solutions impair the wearable technology, with asymmetric stress within the layers induced by the movement of the wearer and impracticalities of having extra layers within an encapsulated textile system. Sheng et al. [17] achieved a single-layer TSC using the fabric itself as the separator, impregnating a central region with a gel electrolyte. However, the cotton substrate had to be 500 μm thick and required a very dense weave. Though this thickness allowed for the separation of the carbon electrodes, later work by the group [18] showed that for multilayer configurations, a thinner fabric achieved much higher capacitances. Clearly, there is a requirement to integrate a separator layer into thinner textiles to enable thinner but more energy-dense TSCs. One method to achieve this is to fill the gaps between the textile fibres/yarns with a porous membrane before the deposition of the electrodes and the impregnation with the electrolyte.

The design of separator membranes is important for all kinds of electrochemical energy storage, from supercapacitors to fuel cells. Important characteristics include chemical stability, ionic conductivity, flexibility, electrolyte update and porosity. Different crystalline and amorphous materials have been evaluated for the fabrication of discrete separator membranes. These include cellulose [19], aluminium oxide nanowire with polyvinyl butyral (PVB) [20], polyacrylonitrile (PAN) [21], polyvinylidene fluoride (PVDF) [22], polyvinyl alcohol (PVA)-tetraethyl orthosilicate [23], poly arylene ether sulfone/polyethylene oxide (PEO) [24] and PVDF-poly methyl methacrylate (PMMA) composite [25]. These materials have a high degree of processability and can be readily fabricated into a porous separator membrane via electrospinning [26] or phase inversion [27]. Phase inversion is a process of immersing a liquid-polymer solution into an anti-solvent solution which removes the original solvent and leaves a porous, solid polymer membrane. However, with the primary research focusing on traditional battery and fuel cell technologies, many of these separators are not suitable for e-textile applications due to poor adhesion to, or integration within, the textile substrate. For example, PAN-based separator membranes have high electrolyte uptake but also have electrolyte leakage issues, PVDF-based separator membranes have high ionic conductivity and high chemical and mechanical resistivity, but their hydrophobicity limits the electrolyte uptake and retention [28]. PMMA-based separator membranes have high porosity and good compatibility with electrolytes due to the polymer’s amorphous nature, but their poor mechanical strength limits their use in energy storage devices [29]. This issue has been improved by mixing and cross-linking different amorphous polymers together with (or without) metal oxide nano/microwires [30]. A separator membrane fabricated in this manner can achieve high porosity, good compatibility with electrolytes and improved mechanical durability suitable for energy storage device design. However, adding these materials to textiles requires additional fabrication processes such as spinning, lamination or embroidery and presents a sub-optimal textile separator design [31]. Therefore, it is essential to develop materials and methodologies that can integrate a membrane layer within a standard textile, utilizing traditional textile fabrication processes. One solution is to introduce polymers such as ethylene-vinyl acetate (EVA) into the separator membrane material matrix to help it fill the space within and adhere to the textile substrate forming a textile and polymer-based separator membrane. EVA is a colourless, flexible and waterproof copolymer of ethylene and vinyl acetate and is a well-known adhesive material for textiles, paper, and other porous materials. The combination of EVA and PMMA creates a mechanically flexible adhesive for optical fibre bonding [32]. The addition of EVA changes the fracture mechanism of the PMMA polymer matrix from crazing to yielding, improving its flexibility [33]. Additionally, numerous fibrils and voids/pores form in the solution-processed EVA/PMMA polymer matrix. These voids also can be used as tunnels and pores for the electrolyte flow, making them suitable as a porous material for separator design.

In this paper, we report a novel approach to the fabrication of carbon-based TSCs, making use of a membrane-impregnated textile substrate. The membrane layer of the proposed supercapacitor was produced via screen printing, followed by a phase inversion process and was composed of a copolymer blend of EVA and PMMA. The effect of polymer ratios on the performance of the membrane is investigated and characterised through microscopy and electrochemical techniques. The reported methodology was shown to be universal across four standard textiles (one polyester, two different polyester-cotton and one silk substrate), with all samples showing promising electrochemical performances and a reduction in process complexity during the electrode deposition stage.

## 2. Materials and Methods

### 2.1. Material

Activated carbon YP-80F was obtained from Kuraray Chemical (Tokyo, Japan), EVA beads (vinyl acetate 25 wt. %), PMMA powder (Mw = 120,000 g/mol), carbon nano-powder additive (<100 nm particle size), 1,2,4-trichlorobenzene, ethanol, dimethyl sulfoxide and Isoamyl acetate were acquired from Sigma-Aldrich (Gillingham, UK). Polyester (denoted as P), two polyester cotton textiles (denoted as PC and PC (thin) and silk (denoted as S) were obtained from Klopman (Frosinone, Italy). The textiles were all plain weaves, with further characteristics presented in Table 1.

### 2.2. Preparation of Textile with a Polymer Membrane and Bending Experiment

Screen printing and phase inversion processes were applied to fabricate the EVA-PMMA copolymer membrane in the cotton textile. The copolymer solution was prepared by mixing EVA beads (vinyl acetate 25 wt. %) with PMMA powder ratios of 1:9, 3:7 and 5:5 by weight, and the textiles with these polymer membranes were denoted as PC1:9, PC3:7 and PC5:5. In total, 2 g polymer blends were dissolved in a solvent mixture of 2.5 mL of 1,2,4-Trichlorobenzene and 2.5 mL of Isoamyl acetate. The polymer blends were completely dissolved in the solvent mixture via heating at 60 °C with a magnetic stirring bar rotated at 600 rpm until the solution became clear. The copolymer solution was then screen-printed twice onto the top of the textile with a screen mesh thickness of 40 µm, printing gap of 0.8 mm, printing pressure of 5.5 kg and speed of 10 mm·s^−1^. After printing, the copolymer-coated cotton was treated with a phase inversion process by immersing the samples in ethanol within a sonication bath for 15 min. The samples were then cured under a vacuum for 2 h at room temperature. After the curing process, some of the copolymer was immersed into the textile and the remaining copolymer formed a membrane layer of about 100 µm on top of the textile.

A group of polyester-cotton textiles with different membranes were bent around a mandrel shown in the Appendix A. These devices were bent around a 3.2 mm diameter mandrel for 500 or 1000 cycles at a cyclical rate of 2 cycles per second.

### 2.3. Fabrication of Supercapacitor Electrodes

The carbon solution was prepared with activated carbon (YP-80F is from Kuraray Chemical), conductive carbon additive and EVA beads (vinyl acetate 25 wt. %) dissolved in a 1,2,4-Trichlorobenzene solvent. The ratio of carbon to binder was 85:15 by weight [34]. The surface of the four textiles was first shaved to reduce pilosity, and the carbon solution was spray coated through a mask on both sides of the textile substrates. The spray nozzle (diameter of 0.3 mm) was positioned 15 cm away from the fabric, and air pressure of 25 psi (1.72 bars) was used for the spraying. Each side of the textile sample was sprayed for 5 s. The mask was removed, and the textile substrate was dried in a fan oven at 120 °C for 10 min. The amount of carbon on the polyester and two polyester-cotton-based devices was 3.56 mg·cm^−2^. For the silk device, the weight of carbon was 1.78 mg·cm^−2^ (maximum possible loading).

### 2.4. Textile Supercapacitor Assembly and Testing

The textile devices were cut into round pieces with a diameter of 1 cm. The organic electrolyte was developed by mixing 1.08 g of Tetraethylammonium tetrafluoroborate salt in 5 mL dimethyl sulfoxide, as demonstrated in a previous paper [35]. The organic mixture was under constant stirring until the mixture became clear. The textile electrode was immersed in the organic electrolyte under vacuum at ~25 mbar for 20 min. For testing, the single-layer textile supercapacitor was placed under compression using spring-loaded grade 303 stainless steel current collectors housed within a Swagelok PFA tube fitting (Swagelok, Solon, OH, USA).

### 2.5. Characterisation and Electrochemical Testing of Flexible Textile Supercapacitor

The device’s surface profile and cross-sectional structure were obtained via scanning electron microscopy (SEM) using a ZES EVO microscope at an operating voltage of 5 kV, 1.68 A with different magnifications. The porosity of the PC textile and PC3:7 textile was obtained by Equation (1) and the distilled water immersion method [36]. Both samples were immersed in the distilled water under vacuum at ~25 mbar for 20 min and then taken out. The excess distilled water on the surface was removed with filter paper, and the content of absorbed distilled water was calculated. The density of the PC textile and PC3:7 were 1.13 and 1.45 g·cm^−3^, respectively.
(1)porosity % =MdρdMdρd+M0ρ0×100%
where *M*_0_ and *ρ*_0_ are the weight and density of the PC textile or PC3:7 before vacuum impregnation. *M_d_* and *ρ_d_* are the weight and density of the distilled water absorbed into the PC textile and PC3:7 textile.

The electrolyte uptake of the PC textile and PC3:7 textile with the proposed membrane were measured via a similar methodology to the porosity test but replaced the distilled water with the proposed electrolyte. The amounts of the electrolyte uptake after the vacuum impregnation process and during the ageing test were determined by Equation (2):(2)uptake % =MeM0×100%
where *M*_0_ is the weight of PC textile or PC3:7 textile with a membrane before vacuum impregnation and *M_e_* is the weight of PC textile or PC3:7 textile with membrane after vacuum impregnation with the electrolyte and after 2, 4, 6, 21 and 24 h left in room temperature at 23 °C. The electrochemical performance of the textile supercapacitor was evaluated using an Autolab PGSTAT101 (Metrohm Autolab, Utrecht, The Netherlands). Electrochemical impedance spectroscopy measurements were performed on a Solartron 1470E Cell Test System (Solartron Analytical, Farnborough, UK) from 250 kHz to 0.1 Hz with an amplitude of 10 mV. The calculations for capacitance, energy and power were based on the GC test result that follows Equations (3)–(5).
(3)C=I×dVdt−1
(4)$$E=0.5\times C\times V_{peak}^{2}$$
(5)P=Et
where *C* is the capacitance, *I* is the cycling current in the GC test, *dV*/*dt* is the rate of change of voltage between 1.44 V (80% *V_peak_*) and 0.72 V (40% *V_peak_*) in the GC test, *E* is the energy stored in the device, *V_peak_* is 1.8 V, *P* is the average power and *t* is the time taken for the supercapacitor to fully discharged.

## 3. Results and Discussion

### 3.1. SEM Imaging

In this work, polyester (denoted as P), two polyester cotton textiles (denoted as PC and PC (thin)) and silk (denoted as S) were used as textile substrates. The copolymer solution was prepared by mixing EVA beads with PMMA powder at ratios (EVA:PMMA) of 1:9, 3:7 and 5:5 by weight, and the textiles with these polymer membranes are denoted as P/PC/PC (thin)/S 1:9, 3:7 and 5:5.

In the EVA-PMMA membrane design, the PMMA contributes toward the porosity of the separator membrane while the EVA enhances the flexibility and adhesion. Higher percentage weights of EVA within the copolymer reduce the shrinkage of the membrane during the phase inversion and thus increase the total coverage of the membrane. However, once the EVA content reaches a critical limit, the membrane loses its porosity and the electrolyte flow through the membrane is restricted. This balancing of the ratio of EVA to PMMA was initially explored across three ratios with the resulting membranes shown in Figure 1a–c, which shows a comparison between samples PC 5:5, PC 3:7 and PC 1:9. The PC 1:9 sample in Figure 1a has less of the membrane layer present in the textile. This is due to the lower mass ratio of EVA within the membrane polymer lattices, resulting in more membrane loss, through shrinkage, during the phase inversion. This shrinkage is uncontrollable and can result in detrimental macro porosity and material loss at the edges of the device. In comparison with the other samples, this membrane is also unlikely to be able to block the electrode material upon deposition, leading to an increased risk of short circuits. The SEM photograph in Figure 1b shows the cross-section view of the sample PC 3:7, where the membrane forms a layer-by-layer structure across the gap between polyester-cotton yarns. As shown in Figure 1c, the PC 5.5 membrane with the lowest percentage of PMMA and the highest percentage of EVA fills the entire area between textile yarns. This will prevent short circuits but will also block the electrolyte flow through the device, reducing the capacitance and increasing the resistance. With the 3:7 ratio demonstrating superior bulk properties, further investigation of this polymer ratio was undertaken. Figure 1d show that the printing and phase inversion process created a porous EVA-PMMA copolymer membrane that adheres to the yarns and fibres of the textile. The membrane itself consisted of polymer lattices with micropores of around 2 µm in diameters (Figure 1e). These micropores are produced uniformly during the phase inversion process and allow for the free movement of the ions within the electrolyte.

Having formed the porous copolymer membrane separator, the next stage in the fabrication process is to deposit the carbon electrodes. For comparison purposes, carbon electrodes were spray coated onto the textiles with and without the membrane in place. Figure 2 show SEM photos of polyester-cotton (PC) and silk (S) textile before and after the spray deposition of the carbon electrode. The SEM photos of the other two textiles (P and PC (thin)) are very similar in appearance to the SEM photos of the PC textile, and they are shown in the Appendix A. The SEM photos in Figure 2a,b show that both the PC and S textiles contain gaps between each fibre. This structure leads to the electrodes contacting each other during the electrode spray coating process, which causes short circuits. This is demonstrated in Figure 2e,f where the SEM photos for spray-coated textiles (PC and S) without the membrane show the carbon layer travelling through the textile structure. This causes carbon material to potentially penetrate through to the other side of the textile causing short-circuits. In comparison, the SEM photos in Figure 2c,d,g,h show that a polymer membrane with an EVA:PMMA ratio of 3:7 overcomes the membrane shrinkage issue observed with the EVA:PMMA ratio of 1:9, creating a homogeneous polymer layer. It not only acts as an electrical separator in the textile supercapacitors but also fills the gap between textile yarns reducing the surface roughness of the textile and blocking any carbon permeation through the substrate. This enhances the carbon deposition and forms a continuous and uniform distributed conductive electrode that leads to a lower electrode resistance and higher power density.

### 3.2. Porosity, Electrolyte Uptake and Ageing Test

Based on the distilled water immersion methodology [33], the porosity of the PC and PC 3:7 samples was evaluated. The PC 3:7 textile demonstrated a porosity of 24.9 ± 6%, which was 6.9% lower than the PC textile only (31.8 ± 4%). The results from the electrolyte uptake and ageing test are shown in Figure 3a. After wetting the samples with an electrolyte containing tetraethylammonium tetrafluoroborate salt and dimethyl sulfoxide (the same electrolyte from the electrochemical testing), the PC 3:7 textile maintained a higher (or comparable when accounting for the uncertainty) volume of electrolyte (51.1 ± 8% mL·cm^−2^) than the PC textile (47.8 ± 10% mL·cm^−2^). The samples were then stored in air at 23 °C for 24 h to assess the electrolyte retention. As can be seen, the PC 3:7 textile exhibited similar or better retention compared to the textile alone throughout ageing. The engineered copolymer membrane, containing EVA and PMMA polymers, fills the gaps between the PC textile yarns and replaces them with much smaller-sized air pores (with this observation agreeing with the SEM results in Section 2.1) and is the reason for the lower porosity. However, despite the lower porosity, the membrane-infused textile was able to absorb a greater amount of electrolyte and retained it better/or equal to the textile. In fact, after 21 h, the retention of the electrolyte was significantly better, suggesting that with further refinement, the membrane could be engineered to retain electrolytes far in excess of the textile alone. The Nyquist plot results of the PC 3:7 textile and PC textile are shown in Figure 3b, with the tests performed over the frequency range of 0.1 to 250 kHz at a 10 mV voltage amplitude. The equivalent series resistance of the PC 3:7 textile (34.6 ± 18.7% Ω or 0.956 ± 16.4% mS·cm^−1^) is more resistive than the PC textile (25.4 ± 15.6% Ω or 1.30 ± 17.5% mS·cm^−1^) (characterised close to 250 kHz) at the crossing point of the real axis. Though higher, the resistance values demonstrate that the membrane does not dramatically influence the equivalent series resistance of textile and thus does not diminish the electrochemical behaviour of the PC textile.

### 3.3. Electrochemical Results

Each device has a surface area of 0.785 cm^2^ with a thickness of approximately 244, 150, 250 and 50 µm for the P, PC (thin), PC and S textiles, respectively. The following results were averaged from five device tests. Cyclic voltammetry (CV) tests were performed at different scan rates between +/−1.8 V at 150 mV·s^−1^, and Galvanostatic cycling (GC) tests were performed between 0 and 1.8 V at a cycling current of 1 mA·cm^−2^.

The CV and GC test results for the single-layer textile supercapacitors based on the impregnated textile substrates (P, PC (thin), PC and S) with an EVA:PMMA ratio of 3:7 are shown in Figure 4a,b, respectively. The single-layer textile supercapacitors with polyester textile (type P), polyester-cotton textile (type PC) and thinner polyester-cotton textile (type PC thin) achieved areal capacitances of 29.2 mF·cm^−2^, 35.6 mF·cm^−2^ and 27.6 mF·cm^−2^, respectively. The single-layer silk textile supercapacitor was shown to work without short circuits and charged to +/−1.8 V; however, it demonstrated a reduced area capacitance of 3.12 mF·cm^−2^. The incorporation of the copolymer membrane as a separator enables supercapacitors to be reliably fabricated in a single layer of silk which cannot be achieved without the membrane in place. However, the nature of the weave with the larger gaps means the resulting capacitor properties are the lowest of the textiles tested in this study.

With the PC textile exhibiting the best electrochemical performance, this textile was further investigated to determine supercapacitor performance with the other membrane ratios. The CV and GC results for the different membrane ratios are shown in Figure 4a,b. These results indicate the single-layer textile supercapacitor based on PC 3:7 achieved a higher capacitance and a lower internal resistance of 320.4 Ω·cm^−2^ (obtained from the voltage drop in GC results) compared to the devices made from the membrane-infused textiles PC 1:9 (925.7 Ω·cm^−2^) and PC 5:5 (1904.1 Ω·cm^−2^). Four out of five devices with the PC1:9 membrane textile short-circuited upon testing. The only successful device demonstrated an areal capacitance of 24 mF·cm^−2^. From the SEM analysis of the 1:9 polymer ratio, this poor reliability is attributed to over shrinkage of the membrane, leading to contact between the electrodes. All five supercapacitors with textile membrane PC5:5 successively passed the CV and GC testing but achieved a much lower capacitance of (11.9 mF·cm^−2^) than the textile membrane PC 1:9 and PC 3:7 supercapacitors (Figure 4c,d). This was because the higher loading of EVA in the membrane blocked the electrolyte flow between carbon electrodes (as theorised from the SEM analysis), reducing its capacitance and increasing its resistance. Whilst previous work demonstrated successful single textile layer supercapacitors fabricated in cotton [17], devices replicating this structure in PC without the membrane effectively failed and could only be charged to 0.6 V under GC cycling (Figure 4d). This demonstrates that single-layer textile supercapacitors cannot be fabricated with this kind of textile without the introduction of a membrane layer, highlighting the significance of this membrane for low-thickness textile supercapacitors.

With the introduction of a secondary layer within the textile, the mechanical stability of the TSC could be impaired. As such, the best performing device (PC 3:7) underwent a series of cyclical bending tests around a 3.2 mm diameter mandrel, with the electrochemical performance being monitored throughout. Figure 5a,b show the CV and GC test results of this single-layer textile supercapacitor based on PC3:7 before bending and after 500 and 100 bending cycles. The CV results in Figure 5a indicate that the supercapacitor’s performance was not negatively influenced by the mechanical bending. As shown in Figure 5b, the discharge time of the device increased slightly, showing a capacitance increase after bending from 35.6 mF·cm^−2^ to 36.2 mF·cm^−2^ after 500 bending cycles and 37.3 mF·cm^−2^ after 1000 bending cycles. The initial IR voltage drop at the beginning of the discharge stage in the GC testing reduced significantly from 0.346 V before bending to 0.285 V after 500 bending cycles and 0.192 V after 1000 bending cycles. This change led to lower device resistance and hence improved maximum power density from 0.253 W·cm^−3^ to 0.448 W·cm^−3^ after 1000 bending cycles. The improvement in supercapacitor properties after bending may be due to the membrane relaxing and becoming fractionally more porous, but any change was too small to be observed in the SEM analysis.

From the results presented in Figure 5c,d, the maximum results for the single-layer textile supercapacitor based on the PC 3:7 textile (tested at a cycling current of 0.25 mA·cm^−2^) are an areal capacitance of 38.2 mF·cm^−2^ or 1.51 F·cm^−3^ (based a device thickness of 250 µm). The maximum energy density and average power density obtained at a cycling current of 0.25 mA·cm^−2^ were 0.68 mWh·cm^−3^/4.88 Wh·kg^−1^ and 0.32 W·cm^−3^/0.057 kW·kg^−1^, respectively. An increase in the cycling current to 8 mA·cm^−2^ results in increases in average power density to 17.1 W·cm^−3^ or 3 kW·kg^−1^; however, the maximum energy density reduces to 0.47 mWh·cm^−3^/3.32 Wh·kg^−1^ and the areal capacitance reduces to 26.4 mF·cm^−2^ or 1.06 F·cm^−3^.

## 4. Conclusions

This paper presented a novel fabrication process for supercapacitors within a single textile layer, regardless of the type, thickness and structure of the fabric. This was delivered through the implementation of a phase-inverted copolymer membrane layer integrated directly within the textile. This membrane layer acts as the electrically insulating layer as well as reducing the permeation of carbon through the substrate during production. This enabled the use of ultrathin textiles as a single-layer substrate for the first time. Three polymer formulations were investigated and shown to produce large variations in both bulk material properties and electrochemical results. The membrane formed from the 3:7 ratio of EVA:PMMA was shown to have the best capacitive, mechanical and physical properties. The lower EVA content (1:9) showed significant shrinkage during the phase inversion and allowed carbon to penetrate the membrane resulting in short circuits. The higher EVA content (5:5 ratio) showed a significantly lower porosity. Electrochemically, the carbon-based TSC produced from the PC3:7 substrate demonstrated the highest areal capacitance, energy and power densities (38.2 mF·cm^2^, 1.51 F·cm^3^, 0.68 mWh·cm^−3^ with an average power density of 0.32 W·cm^−3^). The membrane textile was also shown to be stable under mechanical bending of up to 1000 cycles, indicating a good level of mechanical robustness. Future work will include improving the homogeneity of the membrane and its incorporation into textile-based primary and secondary batteries.

## Figures and Tables

**Figure 1 polymers-14-03399-f001:**
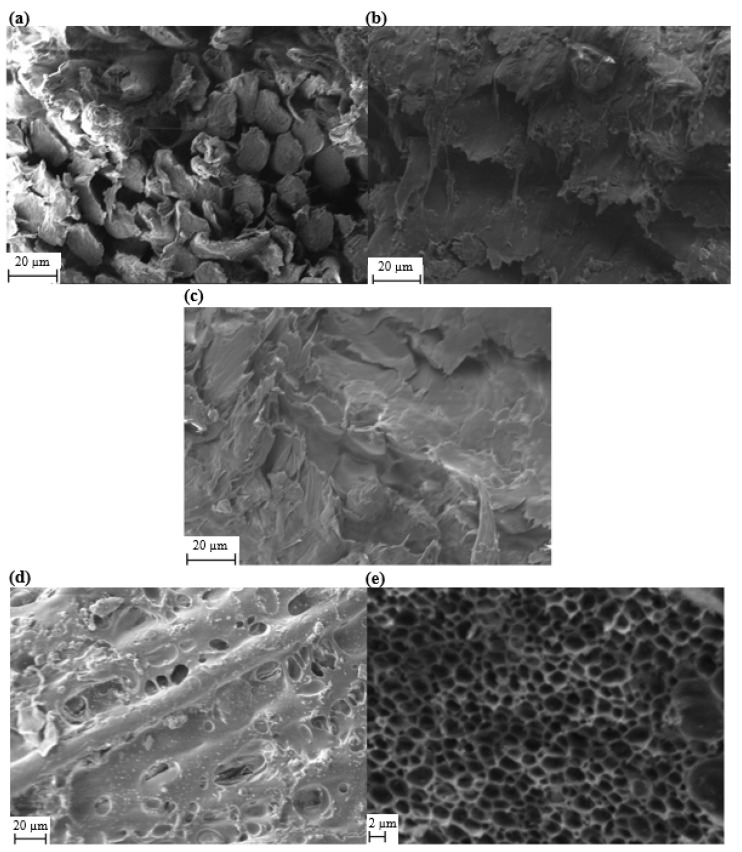
(**a**) Cross-section view SEM photo of sample PC 1:9, (**b**) sample PC 3:7, (**c**) sample PC 5:5, (**d**) plain view SEM photo of sample PC 3:7 and (**e**) high magnification plain view SEM photo of sample PC 3:7.

**Figure 2 polymers-14-03399-f002:**
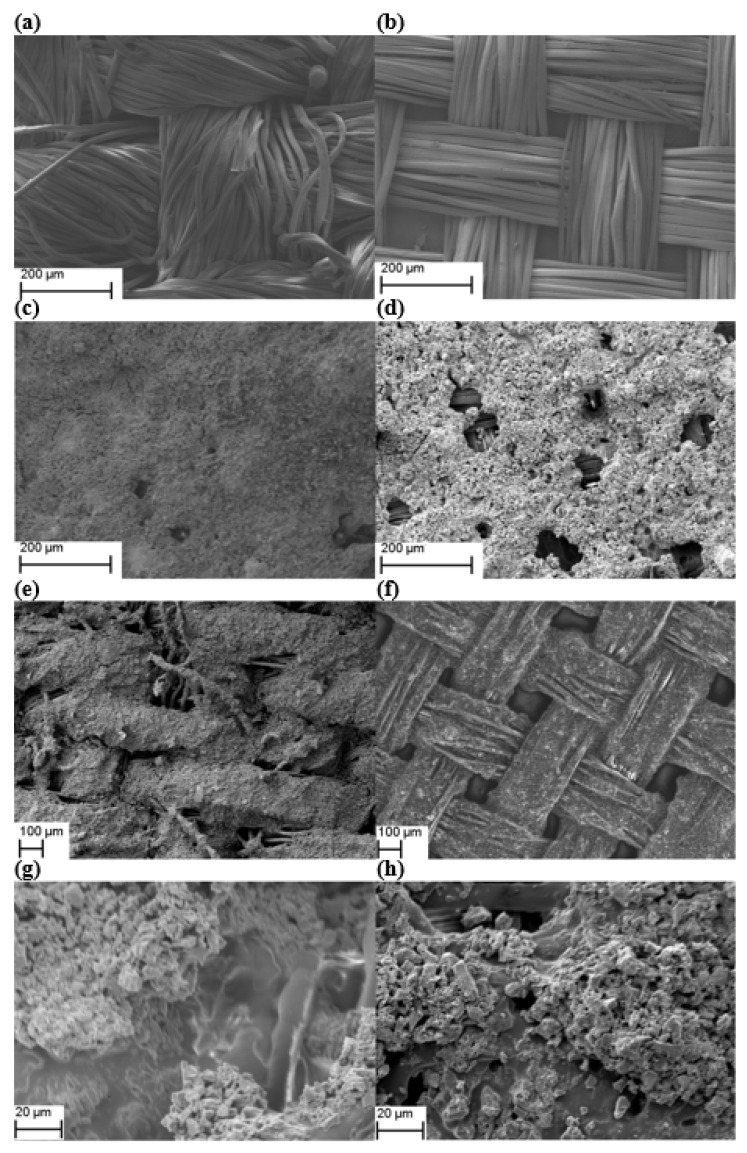
SEM photos showing plan views of (**a**) PC textile, (**b**) S textile, (**c**) PC textile with membrane and carbon coating, (**d**) S textile with membrane and carbon coating, (**e**) PC textile with carbon coating without membrane, (**f**) S textile with carbon coating without membrane, (**g**) PC textile with membrane and carbon coating (high magnification) and (**h**) S textile with membrane and carbon coating (high magnification).

**Figure 3 polymers-14-03399-f003:**
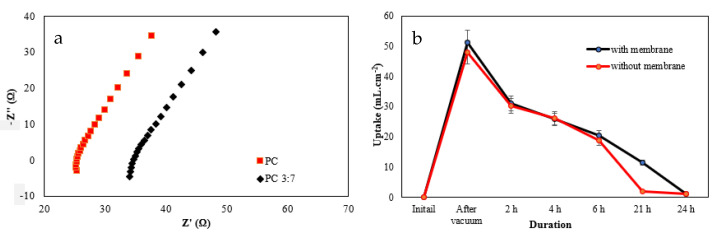
(**a**) Nyquist plots of PC 3:7 textile and PC textile and (**b**) electrolyte uptake and ageing test of PC 3:7 textile and PC textile.

**Figure 4 polymers-14-03399-f004:**
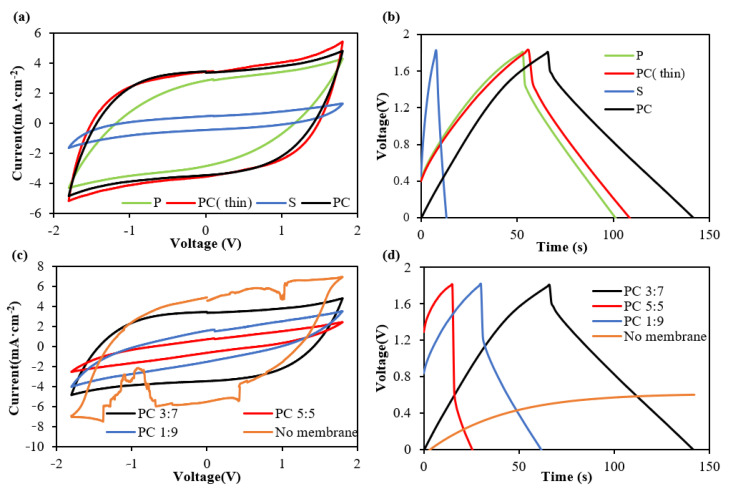
(**a**) CV and (**b**) GC result for single-layer textile supercapacitor based on the different textiles with an EVA:PMMA ratio of 3:7. (**c**) CV and (**d**) GC result for single-layer textile supercapacitor based on PC 1:9, PC 3:7, PC 5:5 and PC without membrane.

**Figure 5 polymers-14-03399-f005:**
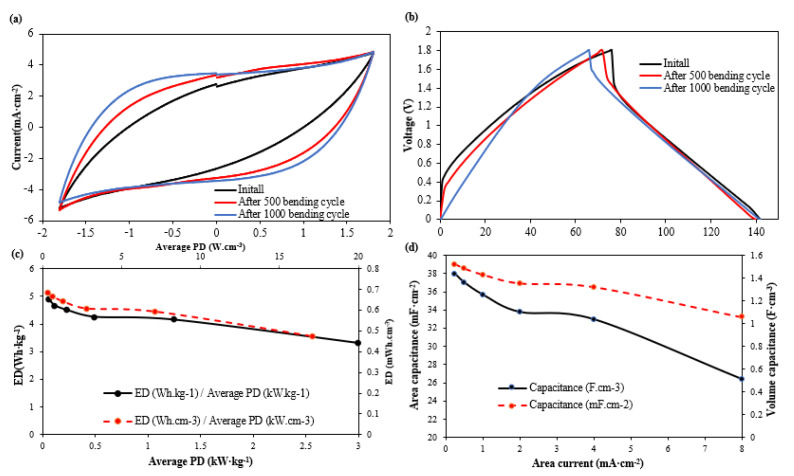
(**a**) CV and (**b**) GC result for single-layer textile supercapacitor based on PC3:7 before and after bending. (**c**) Energy density (ED) and average power density (PD) Ragone plot, (**d**) capacitance variation of single-layer textile supercapacitor based on PC 3:7 and extract from GC test with cycling currents are 0.25, 0.5, 1, 2, 4, 8 mA·cm^−2^. All results are based on device thickness of 250 µm and carbon material loading of 3.56 mg·cm^−2^.

**Table 1 polymers-14-03399-t001:** Characteristics of the textile substrates used throughout this study.

	Polyester-Cotton/Thin (PC Thin)	Polyester-Cotton (PC)	Silk (S)	Polyester (P)
Fibre types	Cotton and Polyester	Cotton and Polyester	Silk	Polyester
Fibre diameter (µm)	12 (Polyester) 15 (Cotton)	12 (Polyester) 15 (Cotton)	~8.3	~20
Ends per inch	16.5	16.5	19.6	13.5
Picks per inch	9.05	9.05	15.7	7.4
Textile original weight (mg·cm^−2^)	16.3	27.1	4.45	23.4
Textile thickness (µm)	150	250	50	244

## Data Availability

The data presented in this study are available on request from the authors.

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
