# Peer review of "Phase-Inverted Copolymer Membrane for the Enhancement of Textile Supercapacitors"

_polymers, 2022, doi:10.3390/polym14163399_

Round 1
Reviewer 1 Report
The authors described a new method to fabricate single layer TSC by using a polymer blend of EVA and PMMA. The fabrication process is well described in details. The authors prepared a series of different polymer mixtures and approparitely designed experiemtns to determine the optimized ratio is 3:7 (EVA:PMMA). The SEM analyses of the structures of fabricated membranes is clear and well coorelated to the performance of fianl devices. The preformace and realibility of these fabricated membranes were adequately characterized by CV and GC tests. The referee recomment to accept this manuscript after considering following comments:
1. The data quaility for the CV test of no membrane coated textile device should be imorived in Figure 4c.
2. The title and author information should also included in the supporting information.
3. Please pay attention to some typos and grammer issues. I do see some of them during reviewing.
Author Response
- The data quaility for the CV test of no membrane coated textile device should be imorived in Figure 4c.
We optimise the sampling rate of the CV test result, improve the quality of Figure 4c and insert the new figure at line 295
- The title and author information should also include in the supporting information.
We add the title and author information in the support information
- Please pay attention to some typos and grammer issues. I do see some of them during reviewing.
We fix the typos and grammar issues of the manuscript; it has been proofread. If the reviewer can be more specific regarding a particular figure or explanation, we would be happy to respond.
Reviewer 2 Report
Current work entitled “Phase inverted co-polymer membrane for the enhancement of textile supercapacitors” by “Yong et al” deliberated on the polymer membranes for the energy storage. Manuscript seems interesting and presented well. The work can be accepted after addressing the following comments.
1. Provide few more keywords
2. In the whole manuscript spacing problem is there at the reference.
3. How the porosity experiments were conducted. Provide elaborative information
4. Scale bar is not clear in the SEM images of figure 1.
5. Explain about the phase inversion processes in the introduction
Author Response
- Provide few more keywords
We add energy storage device, polymer separator and carbon electrode as key words
- In the whole manuscript spacing problem is there at the reference.
We fix the spacing at the reference section to 1.0
- How the porosity experiments were conducted. Provide elaborative information
The detailed information about porosity experiments is presented between line 161 to 165.
- Scale bar is not clear in the SEM images of figure 1.
We improve the scale bar quality of Figure 1 and insert the new Figure at line 197
- Explain about the phase inversion processes in the introduction
We add the explanation of phase inversion processes in the introduction at line 68: Phase inversion is a process of immersing a liquid-polymer solution into an anti-solvent solution which removes the original solvent and leaves a porous, solid polymer membrane